# Performance of the German version of the PARCA-R questionnaire as a developmental screening tool in two-year-old very preterm infants

**Eleonora Picotti[1], Nina Bechtel[2], Beatrice Latal[3], Cristina Borradori-Tolsa[4], Myriam Bickle-Graz[5], Sebastian Grunt[6], Samantha Johnson[7], Dieter Wolke[8], Giancarlo Natalucci[1,3,9]\*, for the Swiss Neonatal Network & Follow-Up Group¶**

1 Department of Neonatology, University Hospital Zurich and University of Zurich, Zurich, Switzerland, 2 Division of Neuropaediatrics and Developmental Medicine, University Children's Hospital Basel, Basel, Switzerland, 3 Child Development Centre, University Children's Hospital Zurich, Zurich, Switzerland, 4 Department of Woman, Child and Adolescent, Geneva University Hospital, Geneva, Switzerland, 5 Department Woman-Mother-Child, University Hospital Lausanne, Lausanne, Switzerland, 6 Division of Neuropaediatrics, Development and Rehabilitation, University Children's Hospital, Inselspital, Bern University Hospital, University of Bern, Bern, Switzerland, 7 Department of Health Sciences, University of Leicester, Leicester, United Kingdom, 8 Department of Psychology, University of Warwick, Coventry, United Kingdom, 9 Larson-Rosenquist Family Foundation Centre for Neurodevelopment, Growth and Nutrition of the Newborn, University of Zurich, Zurich Switzerland

¶ The complete membership of the author group can be found in the Acknowledgments.
\* giancarlo.natalucci@usz.ch

**Data Availability Statement:** The entire dataset is not freely available because of ethical restrictions [the dataset was created by linking national register

## Abstract

### Objective

To validate and test a German version of the revised Parent Report of Children's Abilities questionnaire (PARCA-R).

### Methods

Multicentre cross-sectional study. Parents of infants born <32 gestational weeks, completed the PARCA-R within three weeks before the follow-up assessment of their child at age two years. Infants were assessed using the Mental Development Index (MDI) of the Bayley Scales of Infant Development 2nd edition (BSID-II). Pearson correlation between the Parent Report Composite (PRC) of the PARCA-R and MDI was tested. The optimal PRC cut-off for predicting moderate-to-severe mental delay, defined as MDI<70, was identified through the receiver operating characteristic (ROC) curve.

### Results

PARCA-R and BSID-II data were collected from 154 consecutive infants [51% girls, mean (SD) gestational age 29.0 (2.0) weeks, birth weight 1174 (345) grams] at 23.2 (1.6) months of corrected age. The PRC score [70.5 (31.1)] correlated with the MDI [92.2 (17.3); R = 0.54; p < 0.0001]. The optimal PRC cut-off for identifying mental delay was 44 with 0.81

data and study data (KEK-ZH-Nr. 2012-0278, Ethical Committee of the Zurich University Children's Hospital and Ethical Committee of the Canton Zurich] and confidentiality issues (parents of participants must be asked for authorization to transfer data). However, minimal fully anonymized data set necessary to replicate the study findings are available as supporting information (S1 Minimal fully anonymized data set).

**Funding:** EP was was supported by the Anna Müller Grocholski Foundation (https://neu2017.amg-stiftung.ch). GN was supported by the Swiss National Science Foundation (grant: PZOOP3_161146; http://www.snf.ch). The funders had no role in study design, data collection and analysis, decision to publish, or preparation of the manuscript.

**Competing interests:** The authors have declared that no competing interests exist.

(0.54–0.96) sensitivity (95%-CI), 0.81 (0.74–0.87) specificity, area under the ROC curve of 0.840 (0.729–0.952).

## Conclusion

The German version of the PARCA-R had good validity with the BSID-II and PCR scores < 44 proved optimal discriminatory power for the identification of mental delay at two years of corrected age.

## Introduction

Infants born very preterm (VPT) are a high-risk population for a wide range of neurodevelopmental comorbidities spanning cognitive to motor function and behavioural problems [1]. Early recognition of neurodevelopmental delay through longitudinal follow-up programs is essential for reducing long-term sequelae and promptly introducing supportive interventions [2, 3]. Since 2000, Switzerland has been systematically evaluating the neurodevelopmental achievements of children born VPT by means of standardized face-to-face examinations in accordance with the national recommendation [4]. This practice is also carried out in other countries [5–7]. These gold standard but expensive and time-consuming examinations are performed by trained examiners in each of the 16 specialized centres of the Swiss Neonatal Network and Follow-up Group when VPT children turn 2 and 5 years old. However, 20% of children born at < 30 weeks of gestation and up to 50% of those with gestational age 30 to 31 weeks do not regularly attend the follow up visit (Swiss Neonatal Network ©, data available upon request to the corresponding author). Parental questionnaires may provide a reliable and valid alternative for the assessment of children with an increased risk of developmental problems, such as preterm born babies [8]. These cost-effective and time-saving methods for the investigation of developmental outcomes in domains such as language [9], behavioural [10] and motor [11] function and quality of life [12] may help to increase follow-up rates [13, 14]. However, most of these scales are validated only in English language versions. The Parent Report of Children's Abilities, a parental questionnaire originally developed in the UK [15], was revised and standardized in the UK (PARCA-R) [16] to assess cognitive and language development in preterm born infants aged 24 months [17, 18]. Furthermore, sex- and age-normative data from the test have been recently published making this the only standardized parent completed tool for assessing child development at this age [19]. The questionnaire provides a high test-retest reliability and correlates well with the Mental Development Index (MDI) of the Bayley Scales of Infant Development 2nd (BSID-II) [17, 18] and with the Bayley Scales of Infant and Toddler Development 3rd edition [20, 21], as validated with the Dutch version [22]. Except for a validated Italian version of this tool [23], the questionnaire is not available in any other Swiss national languages. The purpose of the present study was to investigate the performance of a German version of the PARCA-R questionnaire as a reliable and valid developmental screening instrument in the German speaking population of VPT born infants at two years of corrected age in Switzerland.

## Methods

### Study design and population

This was a multicentre cross-sectional study performed at three Swiss tertiary paediatric centres (Basel, Bern & Zurich). Two-year old infants born before 32 weeks of gestational age

between 2010 and 2012 were enrolled for a routine neurodevelopmental follow-up examination at the corrected age of 2 years. Children with congenital malformations or syndromes affecting neurodevelopment, and whose parents did not speak German, were excluded. Neonatal baseline characteristics were extracted from the Swiss Neonatal Network and Follow-up Group's prospective registry of preterm born infants, and defined as previously described [24]. Socioeconomic status was estimated by a validated 12-point score based on maternal education (score 1–6) and paternal occupation (score 1–6), the higher the score the lower the socioeconomic status [25].

## Procedure

The parents of the eligible infants were contacted by telephone by the local follow-up centre that organises the routine neurodevelopmental follow-up assessment at the corrected age of 2 years of their offspring. The parents were asked to complete the German version of the PARCA-R in the three weeks prior to the planned examination, which was sent per mail or given to them directly at their visit to the follow-up centre and completed before the Bayley assessment. Approval to conduct this study was granted by the Ethical Committee of the Zurich University Children's Hospital and by the Ethical Committee of the Canton Zurich (KEK-ZH-Nr. 2012–0278). Written informed consent to the research and the publication of the results was obtained from the parents of each infant.

## Neurodevelopmental measurements at 2 years of corrected age

**Parental questionnaire.**  The Parent Report of Children's Abilities-Revised (PARCA-R) is a standardized assessment of non-verbal cognitive and language development for children aged 24–27 months.

The Parent Report of Children's Abilities-Revised (PARCA-R) is a standardized assessment of non-verbal cognitive and language development for children aged 24–27 months, which typically takes less than 15 minutes for parents (or primary caregivers). No expertise is needed to complete the questionnaire except for a good knowledge of the language used and that the care-giver who completes the PARCA-R has enough time spent with the infant.

Prior to the standardization of the PARCA-R, raw scores for three scales and two sub-scales were computed. These include a 'non-verbal cognition' (range 0 to 34) scale with 34 items on non-verbal cognitive skills, two language development subscales, i.e. 'vocabulary' (range 0 to 100), and a 'sentence complexity' subscale (range 0 to 24), corresponding to the short-form version of the MacArthur Communicative Development Inventories [9]. The sum of the scores of two last subscales produces the "language skills" composite score (range 0–124) which, added to the "non-verbal cognition" score, results in the Parent Report Composite Score (PRC), with values from 0 to 158. In this study, the investigators, who were unaware of the findings of the neurodevelopmental examination, computed all scores according to the instructions from the original survey [16] (all PARCA-R resources are freely available at www.parca-r.info).

The translation of the German Version of the PARCA-R comprised the following steps. The PARCA-R questionnaire was translated from the original English version into German [26] and then translated back to English to ensure the accuracy of the translation by two independent translators who were native speakers of the target language and fluent in the source language (both the English and German version of the questionnaire are freely available from www.parca-r.info). The consistency between the second translation and the original version was analysed and in case of discrepancies, a wording revision of the target version was performed. The pre-final version was reviewed and analysed for cultural characteristics and

proofread by members of the Swiss Follow-up Group to get feedback from clinical experts in the field.

**Neurodevelopmental examination.** The routine neurodevelopmental follow-up examination at the corrected age of 2 years was performed by experienced child neurologists or developmental paediatricians employed at one of the three follow-up centres participating in the study. The examination consisted of a structured neurological assessment and a developmental assessment using the Bayley Scales of Infant Development, 2nd edition (BSID-II) [27]. In this test, the Mental (MDI) and a Psychomotor Developmental Index (PDI) yield age-standardized scores between 49 to 150, with a normative mean (SD) of 100 (15). Infants who were so severely impaired that a structured testing with the BSID-II could not be performed were assigned a Mental Development Index (MDI) and psychomotor development index (PDI) of 49. Vision and hearing were assessed either by direct examination or by caregiver report. Cerebral palsy (CP) was defined [28] and graded [29] according to previously published standards.

## Statistical analysis

Because of the lack of definitive criteria for the determination of the required sample size for psychometric validation studies, a minimum sample size of 100 participants was targeted for the present study, based on previously published general recommendations [30, 31]. Baseline characteristics and follow-up data between participants and non-participants were compared using the independent Student's $t$-, Mann–Whitney $U$ and $\chi^2$ tests as appropriate. Descriptive statistics were used for the findings of the PARCA-R and BSID-II. The MDI of the BSID-II was used as the gold standard to assess the concurrent validity of the PRC of the German version of the PARCA-R, and the Pearson correlation coefficient was computed to assess the relationship between these two scores. Two receiver operating characteristic curves (ROC) were plotted to explore diagnostic accuracy of PRC scores to predict an MDI < 70, i.e. < 2 SD below the normative mean, which defines moderate-to-severe mental delay, and an MDI < 85, i.e. < 1 SD below the mean considered as mild mental delay. The accuracy of the PRC in detecting MDI < 70 and < 85 was measured by computing the area under the ROC. The cutoff points of the PRC with the best predictive performance for an MDI < 70 and < 85 were selected using the Youden's index and sensitivity, specificity, positive predictive and negative predictive values, and their 95% CI were calculated for the corresponding PRC values. The internal consistency of the non-verbal cognition and sentence complexity subscales of the PARCA-R was estimated by means of Cronbach's alpha. The significance threshold was defined as $p < 0.05$, and testing was two-sided. Analyses were performed using SPSS v.25.0 (IBM Corp., Armonk, NY, USA).

## Results

### Study population

Between 2010 and 2012, 765 of 1085 very preterm infants admitted to the neonatal intensive care units of the three participant centres met the eligibility criteria for the study (Fig 1). Of the 554 infants who attended the 2 year follow-up examination, 154 were included in the study as they had both PARCA-R and BSID-II data (51% girls, mean (SD) gestational age 29.0 (2.0) weeks, birth weight 1174 (345) grams). Neonatal and socio-demographic baseline characteristics of the study infants were similar to those of the remaining eligible non-participant infants, except participants had a lower birth weight than non-participants (S1 Table).

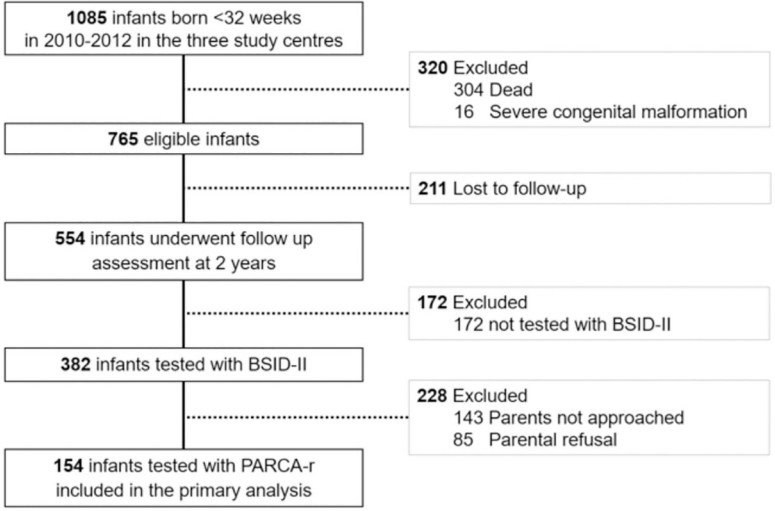

**Fig 1. Study flow sheet.**

## Neurodevelopmental measurements at 2 years of corrected age

Neurodevelopmental assessment of the study infants was performed at a mean (SD) corrected age of 23.2 (1.6) months. The mean (SD) time between completion of the PARCA_R and MDI was 2.7 (8.7) days. The PRC and MDI scores were normally distributed while the scores of the PARCA-R subscales were skewed. The mean (SD) PRC of the study participants was 70.5 (31.1), and the mean MDI 92.2 (17.5), while 31% (N = 48) and 10% (N = 16) had a MDI < 85 and < 70, respectively. Table 1 summarizes the values of the PARCA-R of the study infants, and lists the whole set of findings of the clinical neurodevelopmental examination of the study infants compared to the remaining eligible non-participating infants who were assessed at 2 years of corrected age. No difference in the neurodevelopmental outcome at follow-up was observed between participants and non-participants. The Cronbach's alpha values of the non-verbal cognition and language scales of the PARCA-R were 0.81 and 0.80 respectively.

## Concurrent validity of the German Version of the PARCA-R questionnaire with the BSID-II at 2 years of corrected age

The PRC significantly correlated with the MDI with a Pearson coefficient of 0.54 (p < 0.001) (S2 Table). The correlation between the scores of the PARCA-R regarding the language skills and the MDI (R range 0.53 to 0.54, p > 0.001) was higher than the correlation between the PARCA-r score of non-verbal cognition and the MDI (R = 0.35, p > 0.001).

**Diagnostic utility of the German Version of the PARCA-R questionnaire.** The area (95% CI) under the ROC of the PRC to predict a MDI < 70 (Fig 2) was 0.840 (0.729 to 0.952), and a MDI < 85 (S1 Fig) was 0.774 (0.691 to 0.858) (both p < 0.001). According to the Youden index, the cut-off scores of PRC with the best predictive performance to identify children with MDI < 70 was 44 (i.e. values < 44). With this cut-off, 13 of the 16 infants with MDI < 70 (sensitivity 0.81) were correctly identified as positive, and 112 of 140 infants with MDI ≥ 70 (specificity 0.81) were correctly identified as negative (S2 Fig shows a scatter plot of PRC and MDI values of the 154 study participants). Nineteen per cent of children with a PRC score ≥ 43, i.e. with a normal PARCA-R result, had an MDI < 70, and were thus measured falsely as having no developmental problems, while 20% of children with a PRC value <44, i.e. with abnormal screening result, had an MDI > 70, thus obtaining a false positive. The optimum cut-off PRC

**Table 1. Neurodevelopmental measurements at 2 years of corrected age (PARCA-R screening and clinical examination) of the study participants and of non-participants infants (total n = 554).**

| Mean (SD), n (%) | Participants | Non-participants | P-value |
|---|---|---|---|
| **Age at follow-up, months, mean (SD, range)** | **n = 154** | **n = 228** [a] | |
| | 23.2 (1.6, 21–26) | 23.0 (2.1, 22–28) | 0.399 |
| **PARCA-R, mean (SD, range)** | **n = 154** | | |
| Non-verbal cognition scale | 24.7 (4.2, 3–32) | - | |
| Vocabulary sub-scale | 37.3 (25.5, 0–100) | - | |
| Sentence complexity sub-scale | 9.0 (4.2, 0–23) | - | |
| Language skill composite score | 45.2 (29.1, 0–121) | - | |
| Parent report composite score | 70.5 (31.1, 3–151) | - | |
| **Bayley Scales of Infant Development, 2[nd] Ed.** | **n = 154** | **n = 228** [a] | |
| Mental Development Index, mean (SD, range) | 92.2 (17.5, 49–128) | 92.2 (16.0, 49–132) | 0.974 |
| below 85, n (%) | 47 (31) | 64 (28) | 0.605 |
| below 70, n (%) | 16 (10) | 16 (7) | 0.243 |
| Psychomotor Development Index, mean (SD, range) | 90.7 (16.8, 49–132) | 90.1 (16.8, 49–151) | 0.735 |
| below 85, n (%) | 50 (34) | 76 (34) | 0.939 |
| below 70, n (%) | 15 (10) | 24 (11) | 0.898 |
| **Anthropometric assessment** | **n = 154** | **n = 400** [b] | |
| Weight, mean (SD), kg | 12.9 (10.9) | 13.5 (12.5) | 0.656 |
| z- score, mean (SD) | 0.01 (1.04) | 0.11 (1.16) | 0.388 |
| Length, mean (SD), cm | 86.1 (3.9) | 85.9 (4.7) | 0.767 |
| z- score, mean (SD) | 0.18 (1.16) | 0.21 (1.33) | 0.794 |
| Head circumference, mean (SD), cm | 49.9 (9.0) | 49.8 (8.9) | 0.886 |
| z- score, mean (SD) | -.29 (1.43) | -.38 (1.40) | 0.555 |
| **Neurosensory assessment** | **n = 154** | **n = 400** [b] | |
| Cerebral palsy, n (%) | 6 (4) | 26 (7) | 0.319 |
| GMFCS above 2, n (%) | 0 | 5 (1) | 0.372 |
| Major visual problems, n (%) | 0 | 2 (0.5) | 0.396 |
| Major hearing problems, n (%) | 1 (0.6) | 1 (0.2) | 0.446 |

'SD, standard deviation; IQR, interquartile range

[a], only infants tested with Bayley scales of infant development, 2[nd] edition

[b], all infants visited for the 2 year follow-up examination.

value to identify a MDI < 85 was 63. The predictive values (95%-CI) for the prediction of MDI < 70 and < 85 with the two correspondent cut-offs are listed in Table 2.

## Discussion

This multicentre cross-sectional study provides data on the validity of the German version of the PARCA-R questionnaire as a screening tool identifying delays in cognitive and language development of VPT children at the corrected age of 2 years. The reported optimal cut-off score of the PRC (scores <44) to predict a moderate-to-severe cognitive delay, as defined by a MDI more than 2 SD below the normative mean, was the same as reported in the original British validation study by Johnson and associates [18]. However, the cut-off score provided had slightly lower predictive values in comparison to the UK study. The slightly lower age of the study infant at follow-up in comparison with the UK validation might be partially responsible for the difference in the calculated predictive values. As previously described in other validation studies of the existing English [17, 18, 20], Dutch [22] and Italian [23] versions, all

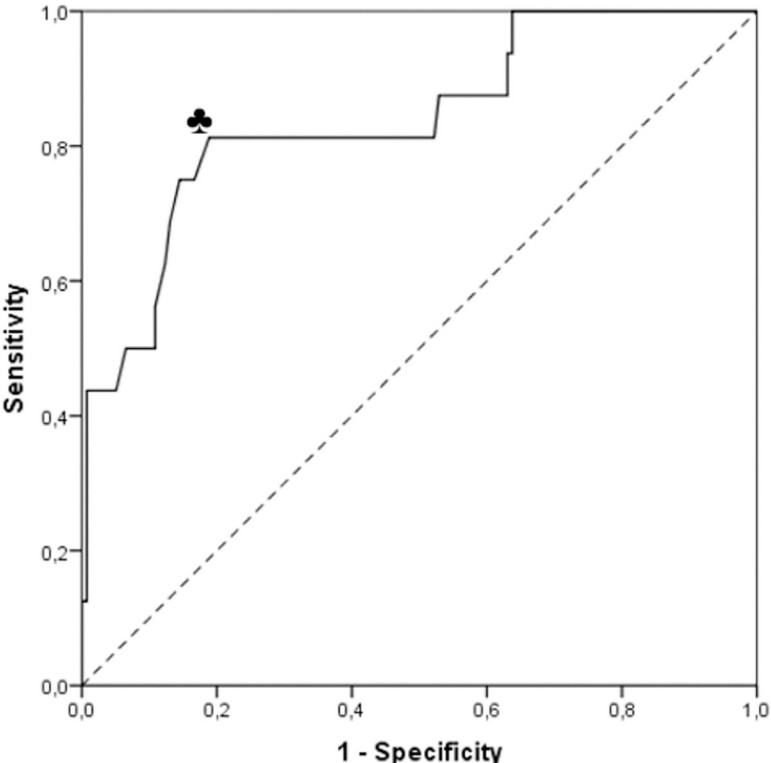

**Fig 2. Receiver operating characteristics curve of prediction of Mental Development Index (Bayley's scales of infant development, 2nd edition) < 70 from Parent Report Composite score (PARCA-r).** ♣ denotes the cut-off score 44 of the Parent Report Composite (PARCA-R) with the best predictive values for identifying infants with a Mental Development Index < 70 (i.e. with mental delay).

subscales and composite scores of the present study correlated moderately with the BSID-II MDI scores with the exception of the non-verbal cognition subscale, which correlated weakly with the MDI. This might be caused by a lack of linearity in the relation between the two types of values, especially in the low range, and by the low variation of the non-verbal cognition sub-scale. The language score contributes more to the PRC than the non-verbal score accounting for 124 points out of 158 in the total PRC score.

While the level of accuracy of the PRC cut-off in screening infants with a severe-to-moder-ate mental delay (i.e. 44) reached a good level, the accuracy of the PRC cut-off with the best

**Table 2. Predictive values of ROC-determined cut-off PRC of 44 and 63 identifying children with MDI below 70 and below 85 at age 2 years, respectively.**

| | Accuracy | False Positive N (%) | False Negative N (%) | Sensitivity (95% CI) | Specificity (95% CI) | Positive predictive value (95% CI) | Negative predictive value (95% CI) | AUC (95% CI) |
|---|---|---|---|---|---|---|---|---|
| **PRC cut-off < 44** | | | | | | | | |
| **MDI < 70** | 0.81 (0.74 to 0.87) | 28 (20%) | 3 (19%) | 0.81 (0.54 to 0.96) | 0.81 (0.74 to 0.87) | 0.33 (0.25 to 0.43) | 0.97 (0.93 to 0.99) | 0.840 (0.729 to 0.952) |
| **PRC cut-off < 64** | | | | | | | | |
| **MDI < 85** | 0.67 (0.59 to 0.75) | 34 (34%) | 14 (30%) | 0.70 (0.55 to 0.83) | 0.66 (0.57 to 0.75) | 0.48 (0.40 to 0.56) | 0.83 (0.76 to 0.89) | 0.774 (0.691 to 0.858) |

PRC, Parent report composite; MDI, Mental development index (< 70, i.e. < -2SD; < 80, i.e. < -1SD); AUC, Area under the ROC (receiver operating characteristic) curve: values ranging 0.90 to 1.00, 0.80 to 0.89, 0.79 to 0.70, and 0.69 to 0.60 indicate excellent, good, fair, and poor accuracy, respectively.

predictive performance for mild delay in the mental development (i.e. 64) was classified as fair. Considering that a neurodevelopmental screening tool should allow the recognition of even infants with mild mental delay, efforts should be done in future to further improve the diagnostic utility of the present German version of PARCA-R.

The study findings support the use of this German version of the PARCA-R (freely available from www.parca-r.info) in the clinical setting and are in agreement with those of previous studies [8, 9, 11, 12, 32], showing that parent reports on the offspring's developmental performances (including the already available validated PARCA-r versions [14, 17–20, 22, 23]) can be considered as a useful screening tool that provides valid and reliable information. By means of the present instrument, infants who have undergone positive screening for moderate to severe mental retardation are identified as being at high developmental risk and should be referred to the follow-up referral centre for a thorough investigation using a clinical examination and a development test.

A strength of this study is the use of a gold standard for the assessment of the mental development of the population studied, the BSID-II [27] which was the developmental test used clinically at the time of the study. In 2012, the Bayley Scales of Infant and Toddler Development, 3rd edition (Bayley-III), was integrated as the new standard in the Follow-up program of VPT infants in Switzerland. As such, this German version of the PARCA-R requires validation against newer editions of the Bayley Scales. Previous studies comparing the English version of the PARCA-R and Bayley-III in other countries have shown that this has good concurrent validity and diagnostic utility [20, 21]. Another strength of the study is the standardized linguistic validation of the German Version of the PARCA-R that included analysis of consistency with the original questionnaire version, analysis for cultural characteristics, and proofreading by clinical experts in the field of child development.

A study limitation is the moderate size of the sample. However, the population studied can be considered as representative of the whole cohort of eligible infants born during the study period and for the group of infants who attended the follow-up examination in terms of both baseline and follow-up characteristics. It is important also to be aware that there are some linguistic differences among the European German-speaking countries. However, the fact that the study participants were only recruited in Switzerland represents a minor limitation. While in the geographic region of the study, the main acquired spoken language of the population is Swiss-German, the official written language form learned at school is standard High German. Additionally, recruitment took place in three urban areas (Bale, Bern, Zurich) with a high rate of German native speakers and the current version of the PARCA-r was specifically designed to be suitable not only for Swiss German speakers, but also for standard high German native speakers coming from regions outside of Switzerland. The third limitation is the fact that families from the lowest socioeconomic status are underrepresented in the present study. With respect to this point, the authors recognize that while proxy questionnaires represent a potential low-cost and time-saving alternative to examiner administered developmental tests, further investigation is still needed to assess whether intellectual and cultural factors could influence the way in which parents from different socio-economic and ethnic backgrounds report their children's skills. However, Johnson and colleagues have previously shown that accuracy of parent report using the PARCA-R was not affected by socio-demographic factors [17].

As the PARCA-R is designed to assess infants' non-verbal cognition and language skills, it is not pertinent to define the lack of information on achievements in other domains, such as motor and behavioural development, as a study limitation. Nevertheless, it is still worth highlighting the fact that this screening tool does not allow for a global developmental screening of preterm born infants. There are some questionnaires that screen motor development at two years of age, however only a few of them were demonstrated to validly and reliably screen

developmental deficits [32] and their availability in multiple validated language versions is limited [33]. There are other questionnaires that focus on the behavioural development at late infancy and early childhood, and are well described in the literature [10]. Since the areas of interest of infant development of these questionnaires fit with the objectives of a structured follow-up for newborns with high neurodevelopmental risk [6], their systematic use should be encouraged in settings where the implementation of follow-up examinations is hampered by lack of hospital resources or parental compliance. These instruments could also be used to systematically monitor developmental outcomes of the growing population of moderately and late premature babies. The PARCA-R is now standardized and produces norm-referenced standard scores (mean 100; SD 15) for cognitive and language development but these are only available for use in the UK population [16, 19]. Future studies could consider obtaining normative data to standardize the German version of the PARCA-R.

In conclusion, in the present sample of VPT children, the German version of the PARCA-R showed good correlation with the results of the BSID-II. The derived PRC cut-off score <44 provides optimal discriminatory power to identify moderate to severe mental delay at 2 years of corrected age. This practical and cost-efficient parental questionnaire may be an alternative for first-line cognitive screening in this high-risk population when direct testing is not possible.

## Supporting information

**S1 Appendix. Members of the Swiss Neonatal Network & Follow-Up Group.**
(PDF)

**S1 Table. Neonatal and socio-demographic baseline characteristics of the study participants and of non-participants eligible infants (total n = 765).** SD, standard deviation; IQR, interquartile range; *, birth weight below 10. percentile; **, above Bell's stage 2.
(PDF)

**S2 Table. Pearson correlation between PARCA-R and Bayley's scales of infant development.** N = 153; R, Pearson R coefficient.
(PDF)

**S1 Fig. Receiver operating characteristics curve of prediction of Mental Development Index (Bayley's scales of infant development) < 85 from Parent Report Composite score (PARCA-r).** ♣ denotes the cut-off score 63 of the Parent Report Composite (PARCA-r) with the best predictive values for identifying infants with a Mental Development Index < 85.
(TIF)

**S2 Fig. Scatter plot of Parent Report Composite and Mental Development Index scores of the 154 study participants.** PRC, Parent Report Composite (PARCA-r); MDI, Mental Development Index, Norm (SD), 100 (15), (Bayley's scales of infant development, 2nd Edition). Vertical line, PRC cut-off score of 44; horizontal line, MDI cut-off score of 70 for defining mental delay.
(TIF)

**S1 Dataset. Minimal fully anonymized data set necessary to replicate the study findings.**
(XLSX)

**S1 Questionnaire.**
(PDF)

## Acknowledgments

We particularly thank the parents who have contributed greatly to this project by consenting to the enrolment of their preterm infant.

The Swiss Neonatal Network and Follow-up Group includes the following local investigators (listed in alphabetical order of study site): Aarau: Cantonal Hospital Aarau, Children's Clinic, Department of Neonatology (Ph. Meyer, C. Anderegg), Department of Neuropaediatrics (A. Capone Mori, D. Kaeppeli); Basel: University Children's Hospital Basel, Department of Neonatology (S. Schulzke), Department of Neuropaediatrics and Developmental Medicine (P. Weber, M. Brotzmann); Bellinzona: San Giovanni Hospital, Department of Pediatrics (G.P. Ramelli, B. Simonetti Goeggel); Berne: University Hospital Berne, Department of Neonatology (M. Nelle), Department of Pediatrics (B. Wagner), Department of Neuropaediatrics (M. Steinlin, S. Grunt); Biel: Development and Pediatric Neurorehabilitation Center (R. Hassink); Chur: Children's Hospital Chur, Department of Neonatology (T. Riedel), Department of Neuropaediatrics (E. Keller, Ch. Killer); Fribourg: Cantonal Hospital Fribourg, Department of Neuropaediatrics (K. Fuhrer); Lausanne: University Hospital (CHUV), Department of Neonatology (J.-F. Tolsa, M. Roth-Kleiner), Department of Child Development (M. Bickle-Graz); Geneva: Department of child and adolescent, University Hospital, Neonatology Units (R. E. Pfister), Division of Development and Growth (P. S. Huppi, C. Borradori-Tolsa); Lucerne: Children's Hospital of Lucerne, Neonatal and Paediatric Intensive Care Unit (M. Stocker), Department of Neuropaediatrics (T. Schmitt-Mechelke, F. Bauder); Lugano: Regional Hospital Lugano, Department of Paediatrics (V. Pezzoli); Muensterlingen: Cantonal Hospital Muensterlingen, Department of Paediatrics (B. Erkert, A. Mueller); Neuchatel: Cantonal Hospital Neuchatel, Department of Paediatrics (M. Ecoffey); St. Gallen: Cantonal Hospital St. Gallen, Department of Neonatology (A. Malzacher), Children's Hospital St. Gallen, Neonatal and Paediatric Intensive Care Unit (J. P. Micallef), Department of Child Development (A. Lang-Dullenkopf); Winterthur: Cantonal Hospital Winterthur, Department of Neonatology (L. Hegi), Social Paediatrics Centre (M. von Rhein); Zurich: University Hospital Zurich (USZ), Department of Neonatology (D. Bassler, R. Arlettaz), University Children's Hospital Zurich, Department of Neonatology (V. Bernet) and Child Development Centre (B. Latal, G. Natalucci). Follow-up Coordinator: Giancarlo.Natalucci@usz.ch.

## Author Contributions

**Conceptualization:** Beatrice Latal, Giancarlo Natalucci.

**Data curation:** Giancarlo Natalucci.

**Formal analysis:** Eleonora Picotti, Cristina Borradori-Tolsa, Myriam Bickle-Graz, Giancarlo Natalucci.

**Funding acquisition:** Beatrice Latal, Giancarlo Natalucci.

**Investigation:** Nina Bechtel, Beatrice Latal, Cristina Borradori-Tolsa, Myriam Bickle-Graz, Sebastian Grunt, Giancarlo Natalucci.

**Methodology:** Beatrice Latal, Samantha Johnson, Giancarlo Natalucci.

**Resources:** Samantha Johnson, Dieter Wolke.

**Supervision:** Myriam Bickle-Graz, Sebastian Grunt, Dieter Wolke, Giancarlo Natalucci.

**Validation:** Samantha Johnson, Dieter Wolke.

**Visualization:** Sebastian Grunt.

**Writing – original draft:** Eleonora Picotti, Giancarlo Natalucci.

**Writing – review & editing:** Nina Bechtel, Beatrice Latal, Cristina Borradori-Tolsa, Myriam Bickle-Graz, Sebastian Grunt, Samantha Johnson, Dieter Wolke.

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
