## [Decision Letter · Decision Letter 0]

15 Apr 2020

PONE-D-20-05014

Performance of the German version of the PARCA-R questionnaire as a developmental screening tool in two-year-old very preterm infants.

PLOS ONE

Dear Dr. Natalucci,

Thank you for submitting your manuscript to PLOS ONE. After careful consideration, we feel that it has merit but does not fully meet PLOS ONE’s publication criteria as it currently stands. Therefore, we invite you to submit a revised version of the manuscript that addresses the points raised during the review process.

We would appreciate receiving your revised manuscript by May 30 2020 11:59PM. To enhance the reproducibility of your results, we recommend that if applicable you deposit your laboratory protocols in protocols.io, where a protocol can be assigned its own identifier (DOI) such that it can be cited independently in the future. For instructions see: http://journals.plos.org/plosone/s/submission-guidelines#loc-laboratory-protocols

We look forward to receiving your revised manuscript.

Kind regards,

Girish Chandra Bhatt, MD, FASN

Academic Editor

PLOS ONE

Journal Requirements:

1. We note that you have indicated that data from this study are available upon request. PLOS only allows data to be available upon request if there are legal or ethical restrictions on sharing data publicly. For information on unacceptable data access restrictions, please see http://journals.plos.org/plosone/s/data-availability#loc-unacceptable-data-access-restrictions.

Reviewers' comments:

Reviewer's Responses to Questions

**Comments to the Author**

1. Is the manuscript technically sound, and do the data support the conclusions?

Reviewer #1: Yes

Reviewer #2: Yes

2. Has the statistical analysis been performed appropriately and rigorously? 

Reviewer #1: Yes

Reviewer #2: Yes

3. Have the authors made all data underlying the findings in their manuscript fully available?

Reviewer #1: Yes

Reviewer #2: Yes

4. Is the manuscript presented in an intelligible fashion and written in standard English?

Reviewer #1: Yes

Reviewer #2: Yes

5. Review Comments to the Author

Reviewer #1: The authors in their multicentric study have validated and tested the performance of a new German version of revised Parent Report of Children's Abilities questionnaire (PARCA-R) against the gold standard Bayley Scales of Infant Development 2nd edition to detect cognitive delay in very preterm born infants at two years of age. They have applied the tests in 154 infants and have concluded that the test had a good correlation with BSID. The cut off score of 44 reliably identifies cognitive delay (BSID- MDI < 70). Cronbach's alpha values were 0.81 and 0.80 for non verbal cognition and language respectively. Indeed. it is a well conducted study, some methodological details would further improve the scientific robustness of the study.

My comments on the manuscript are as follows:-

1. The authors need to provide details of their sample size calculation for the study.

2. What is the average application time for completing this tool and how much expertise is needed to fill the form.

3. Do the authors have data why 85 parents refused to apply the test, is it that they found it difficult to complete and that’s why less educated families were not appropriately represented in the sample.

4. There seems to be a significant delay in submission for publication. The current gold standard is BSID 4th edition. The authors should provide an explanation for the delay. However, they may not write it in the main manuscript.

5. For mild mental delay PARCA-R’s performance is just acceptable (AUC: 0.77, (0.691 to 0.858)). A screening tool should be able to pick milder cases as severe ones will come for medical attention on their own. The tool might need some reason in future to improve upon its properties. The authors must bring this up appropriately in the limitation section of the discussion.

6. Table 1: Were the children with moderate to severe neurodevelopmental impairment (n=21) different from those with psychomotor developmental index / mental developmental index < 70. If not than this needs more detailing.

Reviewer #2: This is an interesting study, though of interest primarily to pediatricians working in German speaking nations. Would suggest addition of the English version of the questionnaire as well, for the benefit of readers who are not familiar with the German language.

6. PLOS authors have the option to publish the peer review history of their article (what does this mean?). If published, this will include your full peer review and any attached files.

Reviewer #1: No

Reviewer #2: Yes: Suvasini Sharma

---

## [Author Response · Author response to Decision Letter 0]

24 May 2020

Editor's comments to the author

Journal requirements. When submitting your revision, we need you to address these additional requirements. Please ensure that your manuscript meets PLOS ONE's style requirements, including those for file naming.

Reply

We checked the revised Manuscript version for the PLOS ONE’s style requirements (including file naming) and correct the format of a subtitle, accordingly (please see page 13, line 209).

Comment 1

We note that you have indicated that data from this study are available upon request. PLOS only allows data to be available upon request if there are legal or ethical restrictions on sharing data publicly.

Reply 1

Please see the statement regarding this point in the rebuttal letter above.

Comment 2

We note that you have included the phrase “data not shown” in your manuscript. Unfortunately, this does not meet our data sharing requirements. 

Reply 2

We understand your point very well and we are sorry that we did not already comment on it in the cover letter of the first submission. We assume that you refer to the sentence on page 4, lines 63-64 (data not published), regarding the low follow-up rate (about 50%) in the subgroup of preterm infants with gestational age 30 to 31 weeks in Switzerland. In fact, data on follow-up rate of preterm born infants are derived from the Swiss Neonatal Dataset of the Swiss Neonatal Network, have never been published, and are only available to the members of the Network. Accordingly, we reworded this part of the sentence hoping that this corresponds to your requirements: (Swiss Neonatal Network ©, data available upon request to the corresponding author) (please see page 4, lines 63-64).

Reviewers’ comments to the author

General reply to comments 1-4 of both Reviewers

We are pleased that the manuscript met the basic criteria required in point 1 to 4 with respect to content and form of the text.

Reviewer 1

Comment 1

The authors need to provide details of their sample size calculation for the study.

Reply 1

We thank Reviewer 1 for this comment. We added a sentence about the sample size calculation in the ‘Statistics’ section, accordingly (please see page 7, lines 150-152). Because of the lack of an evidence based approach for the determination of the required sample size for psychometric validation studies in the literature, we a priori defined a minimum sample size of 100 participants, based on previously published general recommendations (listed as references in the revised manuscript, please see references 30-31).

Please see also Anthoine E, et al. Sample size used to validate a scale: a review of publications on newly-developed patient reported outcomes measures. Health and quality of life outcomes. 2014;12:176 (doi: 10.1186/s12955-014-0176-2).

Comment 2

What is the average application time for completing this tool and how much expertise is needed to fill the form.

Reply 2

The average time for completing the questionnaire is 10-15 minutes. This information is given on page 1 of the questionnaire manual that is freely available at www.parca-r.info (the link is mentioned on page 6 of the main manuscript (lines 120-121). No expertise is needed to complete the questionnaire except for a good knowledge of the language used and that the care-giver who completes the PARCA-R has enough time spent with the infant. In the revised form of the manuscript we add this information, accordingly (please see page 6, lines 113-116).

Comment 3

Do the authors have data why 85 parents refused to apply the test, is it that they found it difficult to complete and that’s why less educated families were not appropriately represented in the sample.

Reply 3

We thank reviewer 1 for this important comment that raises concerns on possible caveats for a general use of the present parental questionnaire on child development. Unfortunately, we do not have a full record of the reasons parent did not participate. The main reason for refusal were the lack of interest in the study and/or of time for participation. In a minority of cases no specific reason for refusal was given at all. No parents gave ‘difficulty in understanding the questionnaire’ as a reason for refusal. Importantly, as refusal was given before sending the questionnaire and only parents who spoke German were approached, non-response due to difficulty in completing the questionnaire can be excluded.

Concerning the second point of comment 3 on education of primary caregivers:

In the ‘Discussion’ (please see pages 18, lines 294-300), we acknowledged that “families from the lowest socioeconomic status were underrepresented in the present study”. However, this limitation regards the whole sample of eligible infants/families (i.e. participants and non-participants). In fact, the socioeconomic status of participants and non-participants (as assessed by a standardized score including the education of the mother and the professional occupation of the father) did not differ. Thus, non-participation was not related to socioeconomic status.

In the same ‘Discussion’ paragraph, we added that “further investigation is still needed to assess whether intellectual and cultural factors may influence the way in which parents from different socio-economic and ethnic backgrounds report their children’s skills”. Nevertheless, it should be noted, that Johnson et al. have previously shown that accuracy of parent report using the PARCA-R was not affected by socio-demographic factors (Dev Med Child Neurol. 2004;46(6):389-97).

Comment 4

There seems to be a significant delay in submission for publication. The current gold standard is BSID 4th edition. The authors should provide an explanation for the delay. 

However, they may not write it in the main manuscript.

Reply 4

We recognize the pertinence of this comment of Reviewer 1. The only reason of the delay in submitting the present study was the lack of resources for analyzing data and in editing the manuscript.

A further factor that explains this delay in minimal part is the fact that new standard versions of development tests, most of which are originally developed in English, come to us with a systematic delay in the translated and validated version (so it was for the transition to Bayley-III and so it will probably be with the transition to Bayley-IV).

Comment 5

For mild mental delay PARCA-R’s performance is just acceptable (AUC: 0.77, (0.691 to 0.858)). A screening tool should be able to pick milder cases as severe ones will come for medical attention on their own. The tool might need some reason in future to improve upon its properties. The authors must bring this up appropriately in the limitation section of the discussion.

Reply 5

We agree with this comment of Reviewer 1 on the reduced diagnostic utility of the present German version of the PARCA-R for recognizing infants with mild mental delay. Accordingly, we reworded the corresponding paragraph that was already present in the ‘Discussion’. As we interpreted this as a limitation of the German Version of the questionnaire and not of the study itself, we opted to maintain the paragraph in its original position (first part of the ‘Discussion’ on the properties of the questionnaire) and not to list it in the limitations’ section (please see page 16, lines 255-260).

Comment 6

Table 1: Were the children with moderate to severe neurodevelopmental impairment (n=21) different from those with psychomotor developmental index / mental developmental index < 70. If not than this needs more detailing.

Reply 6

Among the 21 children with a severe neurodevelopmental impairment (NDI), we included those children with a (mental or psychomotor) developmental index below 70. We acknowledge that we have caused an unfortunate misunderstanding by failing to provide the definition of moderate to severe NDI in the legend of Table 1. This composite outcome is defined as the presence of at least one of following: a developmental (mental or psychomotor) index below 70, cerebral palsy with GMFCS above 2, major visual or hearing problems. The rationale for adding this composite outcome was to provide a summary information regarding the global burden of functional impairment in the study sample.

As this information is redundant and possibly cause of misunderstanding we opted to delete it from table 1 (please see table 1, no change necessary in the main text).

Reviewer 2

Comment 1

This is an interesting study, though of interest primarily to pediatricians working in German speaking nations. Would suggest addition of the English version of the questionnaire as well, for the benefit of readers who are not familiar with the German language.

Reply 1

We thank Reviewer 2 for acknowledging our efforts to offer a developmental screening tool for colleagues working in German speaking nations and for the suggestion for adding the original English version of the questionnaire for the benefit of a large group of readers. Accordingly, we add the link (www.parca-r.info) to download for free the English version of the questionnaire on page 7, line 130 of the revised manuscript free of charge. Readers can also use this link to download for free other language versions of the questionnaire.

---

## [Decision Letter · Decision Letter 1]

6 Jul 2020

Performance of the German version of the PARCA-R questionnaire as a developmental screening tool in two-year-old very preterm infants.

PONE-D-20-05014R1

Dear Dr. Natalucci,

We’re pleased to inform you that your manuscript has been judged scientifically suitable for publication and will be formally accepted for publication once it meets all outstanding technical requirements.

Kind regards,

Girish Chandra Bhatt, MD, FASN

Academic Editor

PLOS ONE

Additional Editor Comments (optional):

Reviewers' comments:

Reviewer's Responses to Questions

**Comments to the Author**

1. If the authors have adequately addressed your comments raised in a previous round of review and you feel that this manuscript is now acceptable for publication, you may indicate that here to bypass the “Comments to the Author” section, enter your conflict of interest statement in the “Confidential to Editor” section, and submit your "Accept" recommendation.

Reviewer #1: All comments have been addressed

2. Is the manuscript technically sound, and do the data support the conclusions?

Reviewer #1: Yes

3. Has the statistical analysis been performed appropriately and rigorously? 

Reviewer #1: Yes

4. Have the authors made all data underlying the findings in their manuscript fully available?

Reviewer #1: No

5. Is the manuscript presented in an intelligible fashion and written in standard English?

Reviewer #1: Yes

6. Review Comments to the Author

Reviewer #1: (No Response)

7. PLOS authors have the option to publish the peer review history of their article (what does this mean?). If published, this will include your full peer review and any attached files.

Reviewer #1: **Yes: **Prashant Jauhari

---

## [Editor Report · Acceptance letter]

24 Aug 2020

PONE-D-20-05014R1 

Performance of the German version of the PARCA-R questionnaire as a developmental screening tool in two-year-old very preterm infants. 

Dear Dr. Natalucci:

I'm pleased to inform you that your manuscript has been deemed suitable for publication in PLOS ONE. Congratulations! Your manuscript is now with our production department. 

Kind regards, 

on behalf of

Dr. Girish Chandra Bhatt 

Academic Editor

PLOS ONE